# Study protocol: the ear–nose–throat (ENT) prospective international cohort of patients with primary ciliary dyskinesia (EPIC-PCD)

Myrofora Goutaki [1,2] Yin Ting Lam [1] Mihaela Alexandru,[3,4] Andreas Anagiotos,[5] Miguel Armengot,[6,7] Emilie Bequignon,[8] Mieke Boon,[9] Andrea Burgess,[10] Andre Coste,[8,11] Nagehan Emiralioglu,[12] Ela Erdem,[13] Eric G Haarman,[14] Amanda Harris,[15] Sara-Lynn Hool,[16] Bulent Karadag,[13] Sookyung Kim,[3] Philipp Latzin,[2] Natalie Lorent,[17] Ugur Ozcelik,[12] Ana Reula,[18,19] Jobst Roehmel,[20] Christine van Gogh,[21] Panayiotis Yiallouros,[22,23] Soeren Marian Zappe,[24] On behalf of the EPIC-PCD team, Jean Francois Papon[3,4]

For numbered affiliations see end of article.

**Correspondence to**
Dr Myrofora Goutaki;
myrofora.goutaki@ispm.unibe.ch

## ABSTRACT

**Introduction** Primary ciliary dyskinesia (PCD) is a rare, genetic, multiorgan disease with an estimated prevalence of 1 in 10 000. It affects mainly the upper and lower airways due to impaired mucociliary clearance. Almost all patients have sinonasal or otologic (ear–nose–throat, ENT) problems, although the ENT clinical phenotype may present great variability. Despite that, data on PCD ENT manifestations are scarce and based on small single-centre studies. To date, we know little about the spectrum and severity of PCD ENT disease, its association with lung disease, its course over life and its determinants of prognosis.

This study protocol describes the aims and methods of the first prospective, observational, multinational cohort study focusing on ENT disease in patients with PCD.

**Methods and analysis** The ENT prospective international cohort of patients with PCD (EPIC-PCD) is a prospective standardised observational clinical cohort set up as a multinational multicentre study, embedded into routine patient care. It aims to longitudinally characterise ENT disease in patients with PCD and its association with lung disease, and to identify determinants of its prognosis. Patients of all ages, diagnosed with PCD who undergo an ENT clinical assessment at least once a year at one of the participating centres will be invited to participate. Collected data include diagnostic test results, results of ENT examinations, lung function measurements, information on management of ENT disease and patient-reported data on clinical symptoms and health-related quality of life (QoL). Data are collected using the standardised PCD-specific FOLLOW-PCD form and the validated QoL-PCD questionnaire.

**Ethics and dissemination** The study has been reviewed and approved by the Human Research Ethics Committees at all participating centres, based on local legislation. The results of the study will be published in scientific journals, presented at scientific conferences and disseminated to participants and national patient organisations.

**Trial registration** NCT04611516.

## Strengths and limitations of this study

► The ear–nose–throat prospective international cohort of patients with primary ciliary dyskinesia (EPIC-PCD) is the first prospective international multicentre clinical study focusing on upper airways disease in patients with PCD.

► The study is embedded in routine patient care, which allows recruitment of large numbers of patients, necessary in rare diseases research.

► The study combines objective assessments of the upper airways and patient-reported information on symptoms and health-related quality of life (QoL), all collected with disease-specific tools, in a standardised way.

► The cohort aims to be representative of paediatric and adult patients with PCD, followed up at established PCD clinical centres; however, it will not represent patients with atypical symptoms who remain undiagnosed or live in areas where diagnosis and care of PCD is not well established.

► The EPIC-PCD has the potential to become a valuable research resource in order to help understand better the disease and to improve diagnosis, clinical care and QoL of patients with PCD.

## INTRODUCTION

Primary ciliary dyskinesia (PCD) is a rare genetic, multiorgan disorder affecting approximately 1:10 000 individuals.[1 2] Mutations in disease-causing genes result in defects in ciliary structure or function, leading to a wide range of symptoms from different organ systems.[3–5] The clinical phenotype of the disease is heterogeneous; however, reduced airway mucociliary clearance predisposes almost all affected individuals to recurrent upper and lower respiratory infections.[6 7]

Persistent symptoms from the upper airways and ears (ear–nose–throat: ENT) are a common characteristic of most patients with PCD. Ear symptoms commonly present in children with PCD as severe and bilateral otitis media with effusion (OME).[8–11] OME usually resolves spontaneously in otherwise healthy children, but in PCD it seems to persist longer and might need specific management.[8 12–15] In some cases, persistent OME may lead to hearing impairment and cause delays in speech development.[16 17] A newly published study evaluated 47 children in two PCD centres in North America and found that 52% of children with PCD-related hearing loss were not aware of their hearing deficit that was present on audiological assessment.[18] While otologic involvement seems to improve with age in patients with PCD, sinonasal disease appears to persist and may worsen in adulthood.[19] First manifestations appear sometimes soon after birth and are usually non-specific in the form of rhinitis and episodic facial pain due to recurrent infections leading gradually to chronic rhinosinusitis (CRS).[20–24] In adults, a recent study identified sinonasal disease as a primary cause of decreased quality of life (QoL) in patients with PCD.[19] Hypoplasia and agenesis of paranasal sinuses have also been described, while nasal polyps are more common in patients with PCD compared with the general population, although this complication has not been well characterised.[19 25 26]

Despite being so frequent and impairing patients' physical health and QoL, data on ENT disease in PCD are still scarce. Several large national and multinational research collaborations have made important advances in PCD research in the recent years, though most clinical studies focus on improving diagnosis and characterising lung disease.[27–30] Only few retrospective and single-centre studies have described ENT manifestations in some details. Most studies focused on children and little is known about age-related changes,[8 13] while studies, including standardised sinonasal and otologic evaluations, were rare. We lack knowledge about the spectrum and severity of ENT disease in patients with PCD, its association with lung disease, its course over life and determinants of prognosis.

This study protocol describes the aims and methods of the first prospective, observational, multinational cohort study focusing on ENT disease in patients with PCD.

## Study objectives

This study aims to characterise ENT disease in patients with PCD and its association with lung disease, and to identify determinants of its course and prognosis. Specifically, we aim to: (a) assess the prevalence and severity of sinonasal and otologic symptoms and the frequency and range of signs and physiological findings assessed during standardised ENT physical examination, and describe possible differences by age; (b) study the association of sinonasal and otologic disease with lung disease in patients with PCD and (c) identify determinants of the course and prognosis of sinonasal and otologic disease in patients with PCD.

## METHODS AND ANALYSIS

### Study design

The ENT prospective international cohort of patients with PCD (EPIC-PCD) is a prospective observational clinical cohort set up as a multinational multicentre study. It is embedded into routine patient care of participating reference centres for PCD and patients continue to be managed according to local procedures and protocols. The study is hosted at the Institute of Social and Preventive Medicine (EPIC-PCD data centre) at the University of Bern, Switzerland, and is managed in collaboration with participating centres (EPIC-PCD clinics).

Patient data are collected prospectively. Baseline information is collected at the enrolment of the patient in the study and follow-up data during clinical follow-up visits at the participating centres. The study has no foreseen endpoint. Patient follow-up is planned for a minimum period of 4 years; however, it will be extended if collaborators agree.

### Study population

#### Inclusion criteria and participating centres

Patients of all ages, diagnosed with PCD, according to the European Respiratory Society (ERS) guidelines, and followed-up at one of the participating centres, are invited to participate in the EPIC-PCD.[27] Patients are excluded if they do not undergo an ENT clinical assessment at baseline and at least once a year during the study period.

Patient recruitment has started at the following centres (EPIC-PCD clinics; figure 1):
- University Hospital Bicêtre Paris-Sud, France.
- Centre Hospitalier Intercommunal Creteil and University Hospital Henri Mondor, France.
- Hospital Universitario La Fe in Valencia, Spain.
- University Hospital of Southampton, UK.
- University Hospital of Leuven, Belgium.
- VU University Medical Center in Amsterdam, the Netherlands.
- University of Cyprus;
- Hacettepe University in Ankara, Turkey;
- Marmara University in Istanbul, Turkey;
- University Children's Hospital Charité-Universitätsmedizin in Berlin, Germany.
- University Children's Hospital in Bern, Switzerland.

After the establishment of the study in these initial EPIC-PCD clinics, we will invite more PCD centres to join the study and begin patient recruitment.

### Study procedures

#### Patient identification and recruitment

Eligible patients are identified and invited to participate in the study, and receive all study information by the local clinical team at the EPIC-PCD clinics. Patients are recruited during a clinical follow-up visit; there might

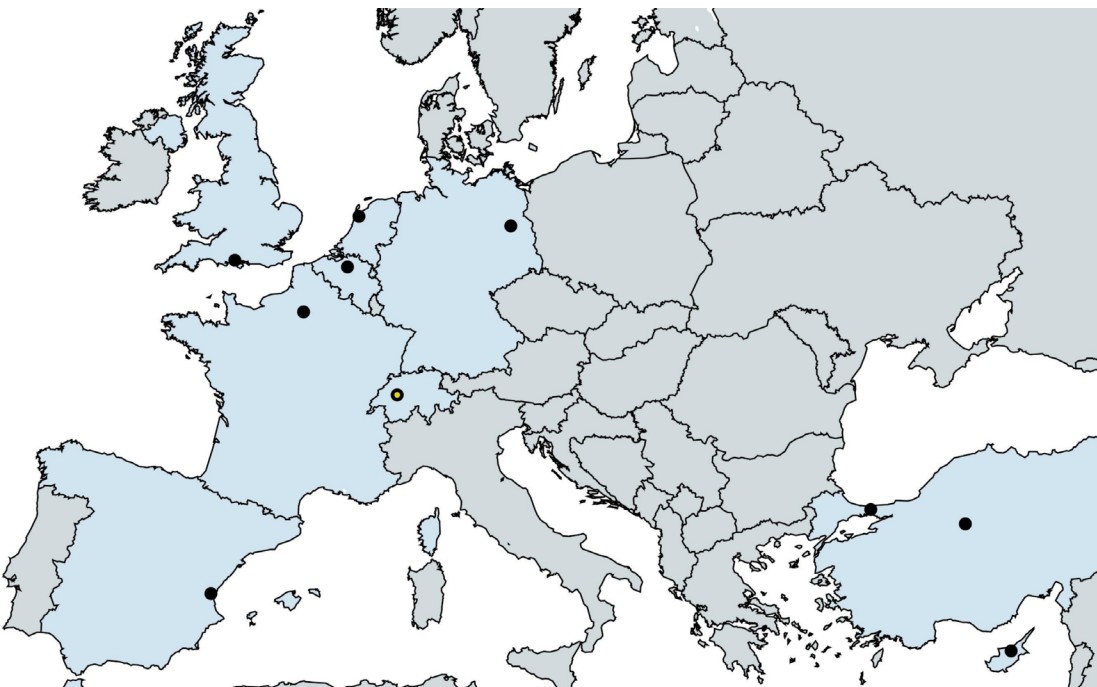

**Figure 1** Centres currently participating to the EPIC-PCD cohort study. The EPIC-PCD clinics are marked with a black dot and the EPIC-PCD data centre with a yellow dot. EPIC-PCD, ear–nose–throat prospective international cohort of patients with primary ciliary dyskinesia.

be small variations in recruitment and data collection between participating centres to allow the study to be embedded in the regular clinical routine.

### Clinical assessments

Eligible patients are followed regularly at the EPIC-PCD clinics, at 3–6-month intervals, as planned for clinical reasons, regardless of their participation in the study. Each patient undergoes a detailed sinonasal and otologic examination by ENT specialists, at minimum once a year, during a scheduled follow-up visit. Additional ENT examinations are performed on clinical indication in-between follow-up visits. Patients will not be subjected to additional invasive measurements solely for the purposes of the study; examinations and tests are only done if clinically indicated or as part of the local follow-up protocol. All examinations and tests are performed following the local standard operating procedures of the clinic. The extended study team discusses differences and ensures that collected data are standardised and comparable.

### Data collection instruments

For the collection of patient data, we use standardised PCD-specific instruments, specifically FOLLOW-PCD and QoL-PCD.

### FOLLOW-PCD

It is a disease-specific form for standardised prospective data collection during routine clinical follow-up of patients with PCD.[31] It was developed by a multidisciplinary, international working group of the better experimental approaches to treat PCD (BEAT-PCD) COST action network.[28 32] FOLLOW-PCD has a modular

structure, which permits different members of the clinical team (pulmonologists, ENT physicians, diagnostic experts, physiotherapists, lung function technicians, etc) to complete different modules. FOLLOW-PCD consists of seven modules that enable an extensive evaluation at baseline and annual reviews, when all modules are completed, and a shorter evaluation at routine (eg, 3 monthly) follow-up, where only a few modules are used. For the EPIC-PCD study, we use specific modules to collect clinical data relevant to the study, particularly demographic and diagnostic information, results of the physical ENT examination, results of clinical tests and data on hospitalisations and treatments related to the upper airways.

Some modules of FOLLOW-PCD have been developed into patient questionnaires (FOLLOW-PCD questionnaire) that can be completed by the patients during each clinical follow-up visit. The FOLLOW-PCD questionnaire includes detailed questions on frequency and severity of upper and lower respiratory symptoms and health-related behaviours (eg, smoking and exercise).[31] There are three age-specific versions of the questionnaires: one for adult patients, one for adolescents 14–18 years old and one for the parents or caretakers of patients aged less than 14 years. The FOLLOW-PCD questionnaire has been originally developed in English, German and Greek and is translated in additional languages using a standardised procedure.

In addition to the FOLLOW-PCD questionnaire, we will ask the patients to complete once a year the QoL-PCD, a disease-specific, health-related QoL instrument for PCD, validated and translated in several languages.[33–36] There are four different versions available for (a) children

| BASELINE visit (patient recruitment) | FOLLOW-UP visits (at least once/year ) |
|---|---|
| **MINIMUM REQUIREMENTS**<br>• FOLLOW-PCD patient questionnaire<br>• Standardised ENT examination including:<br>  Audiometry<br>  Tympanometry<br>  Nasal endoscopy<br>• Spirometry<br>• Height/Weight<br>• Information on ENT management<br><br>• Diagnostic information<br>• Baseline medical history | • FOLLOW-PCD patient questionnaire<br>• Standardised ENT examination including:<br>  Audiometry<br>  Tympanometry<br>  Nasal endoscopy<br>• Spirometry<br>• Height/Weight<br>• Information on ENT management |
| **IF AVAILABLE**<br>• QoL-PCD questionnaire/ SNOT-22<br>• Additional ENT examination results e.g.:<br>Nasal function tests/ Audiovestibular function tests/<br>Auditory brainstem evoked response/ Sinus, ear CT<br>• Plethysmography, Multiple breath washout<br>• Microbiology test results from upper and lower airways | • QoL-PCD questionnaire/ SNOT-22<br>• Additional ENT examination results e.g.:<br>Nasal function tests/ Audiovestibular function tests/<br>Auditory brainstem evoked response / Sinus, ear CT<br>• Plethysmography, Multiple breath washout<br>• Microbiology test results from upper and lower airways |

**Figure 2** Overview of the information collected during the EPIC-PCD cohort study. ENT, ear–nose–throat; EPIC-PCD, ear–nose–throat prospective international cohort of patients with primary ciliary dyskinesia; SNOT-22, Sino-Nasal Outcome Test-22.

(6–12 years), (b) adolescents (13–17 years), (c) adults (≥18 years) and (d) a parent-proxy measure for children ≤12 years. The different versions of the questionnaire are divided into 7–10 hypothesised scales generated by multi-trait analysis. All versions include two scales about upper respiratory symptoms, and ears and hearing symptoms, which are particularly relevant to this project. The QoL-PCD has demonstrated good internal consistency and test–retest reliability, and is a valuable tool for observational and interventional studies in PCD.

The FOLLOW-PCD questionnaire does not address QoL; therefore, the two questionnaires provide important complimentary information. Both questionnaires have been translated in the local language of each participating EPIC-PCD clinics.

Lastly, some centres use as part of their regular patient follow-up the Sino-Nasal Outcome Test-22 (SNOT-22) questionnaire that assess sinusitis-related QoL.[37] In these centres, we will also record the SNOT-22 score at baseline and at follow-up for all participating patients.

### What information is collected

Figure 2 provides a brief overview of the information collected from participating patients at the baseline and follow-up visits, including the minimum required data, which are assessed for all patients in all participating centres and additional data from tests performed in few centres and which will be collected if available for subgroup analysis to address additional research questions. Small exemptions are allowed when a minimally required examination was not performed due to, for example, the patient's very young age, as long as each patient undergoes a specialised ENT examination at least once a year.

### ENT examinations

The ENT examinations performed in the participating centres in addition to physical examination include nasal endoscopy (either rigid or flexible), nasal function tests (eg, patency and olfaction) tests, tympanometry, audiometry and auditory brainstem evoked response. Nasal endoscopy, tympanometry and audiometry are performed regularly at all centres during follow-up and belong to the minimum requirements for participation to the study; the remaining tests are performed only in selected, specialised centres either at initial diagnosis or during follow-up. When clinically indicated, patients will undergo CT of the sinuses or the ears. When available, we will collect data of previous imaging examinations of the sinuses or the ears from clinical records.

The data collected during the ENT examinations are predefined based on the ENT examinations module of the FOLLOW-PCD standardised form.[31] Nasal endoscopy data include the presence and consistency of nasal discharge (eg, serous and mucopurulent), evaluation of nasal mucosa (eg, presence of erythema or oedema), visual evaluation of nasal polyps using the Lildholdt score,[38] evaluation of hypertrophy or atrophy of nasal turbinates and description of septum and any deformities caused due to chronic infection. Otoscopy data include the presence and consistency of ear discharge, visual description of the tympanic membrane (eg, tympanic perforation, retracted membrane and tympanic sclerosis) and the presence of acute otitis media or OME. Results of tympanometry are recorded using the description of tympanogram type by Jerger and of audiometry using the type of audiometry and the WHO hearing loss grades.[39] In addition, we record the presence of grommets or use of hearing aids. When sinus imaging is performed, we record the following findings: the presence of aplasia,

hypoplasia, thickening of the bone and Lund-Mackay score.[40]

### Other data from clinical records

At recruitment, baseline medical information and results of diagnostic tests will be extracted from clinical records and recorded using FOLLOW-PCD. In addition to the medical data from the upper airways, we collect data from lung function measurements, including spirometry and when available multiple breath-washout measurements and lung imaging. This information will allow us to assess associations between upper and lower respiratory disease in PCD.

On indication, nasopharyngeal or nasal swab samples, and when possible middle meatal samples, and sputum or cough samples for microbiological cultures and results will be collected. In case of otorrhoea, ear drainage samples will also be taken. We record the type of sample, the isolated pathogens and antibiotic resistance. We also collect information related to hospitalisations and surgical procedures related to upper airways disease and data on upper airway management.

### Information from questionnaires

Participating patients are asked to complete the FOLLOW-PCD questionnaire at every visit and the QoL-PCD questionnaire at least once a year, as described earlier. The FOLLOW-PCD questionnaire includes sections on upper and lower respiratory symptoms, health-related behaviours (eg, exercise and smoking) and environmental factors. The QoL-PCD questionnaire assesses how the disease affects the QoL of patients with PCD.

### Study database

All collected data are entered into a secure web-based database using the database programme, Research Electronic Data Capture (REDCap) hosted by the Clinical Trials Unit, Bern.[41] REDCap is widely used in academic research and allows data entry and extraction in various formats. Contributing centres are responsible for the pseudonymisation of the data contributed to the study. Only limited research staff from the EPIC-PCD data team has access to the whole dataset; local investigators at the EPIC-PCD clinics have access only to data from their clinic (box 1). Data generation, transmission, storage and analysis of health-related personal data within the study follow the Swiss legal requirements for data protection. All study team members are trained on data protection and dealing with particularly sensitive data.

### Study power and sample size

We expect to recruit at least 450–550 patients in EPIC-PCD. This number is based on the study feasibility and the number of patients followed-up in the EPIC-PCD clinics and is considered large enough for a prospective study in such a rare disease and much larger than any other available PCD dataset on ENT disease. After the establishment of the cohort, it is now possible for more centres to join and contribute patients, resulting in even higher

---

**Box 1  Contributing to and accessing data in the EPIC-PCD**

**How to contribute data**

► Centres that wish to participate in the project and contribute data can contact the EPIC-PCD data team. As soon as ethical approval is obtained and centre representatives have signed a data delivery agreement, they will receive a password to access the online software REDCap and they will be able to enter their data directly prospectively. We offer information and technical support wherever needed.

**How to access data**

► Participating centres keep constant access to their datasets and can export them directly in various formats for local analyses. Participating centres may also extract their datasets and contribute relevant data to other multinational projects or to the international PCD registry.

► The use of EPIC-PCD data from several centres for studies and publications is regulated by the EPIC-PCD committee, which consists of members of the EPIC-PCD data centre and representatives of the EPIC-PCD clinics. A series of publications are originally planned to be led by the EPIC-PCD data team, as part of the research grant funding the set-up and management of the study.

► For additional publications, collaborators should submit a concept sheet describing the planned analysis and publication to the committee, which will approve or oppose the planned publication and use of data. If the planned analysis is approved, a publication agreement will be signed and the EPIC-PCD data centre will prepare a partial dataset for the proposed analysis and will work closely with the lead researchers of each planned study for methodological input and support.

► For further details, contact: myrofora.goutaki@ispm.unibe.ch.

EPIC-PCD, ear–nose–throat prospective international cohort of patients with PCD; REDCap, Research Electronic Data Capture.

---

numbers of patients. The planned analyses are up to a point exploratory and do not follow specific hypotheses, although we have few assumptions based on the limited literature. This does not allow for a precise power calculation; however, we will ensure that each performed analysis has sufficient power to draw meaningful conclusions.

### Planned analysis and outcomes of interest

First, we will describe prevalence and severity of ENT problems, stratified by age, age at diagnosis, sex, country, ciliary ultrastructural defect or genetic mutation. Outcomes of interest will be sinonasal and otologic symptoms and signs (eg, rhinitis, loss of smell, sinusitis, otitis and impaired hearing) and the results of ENT examinations (mainly of nasal endoscopy, tympanometry, audiometry and when available imaging and SNOT-22). We will estimate prevalence of binary outcomes and mean values of continuous outcomes by using constant-only models. We will use logistic regression models for binary outcomes, and linear models for continuous outcomes. Associations with determinants will be assessed in multivariable mixed models.

As a second step, we will assess the association of baseline ENT problems with lung disease progression expressed

in terms of lung function tests, sputum cultures and, if available, lung imaging results. For this analysis, lung function will be the main outcome, measured primarily with spirometry and plethysmography measuring forced expiratory vital capacity and forced expiratory volume in 1 s. Data will be converted into z-scores adjusted for sex, height, age and ethnicity, using the Global Lung Initiative reference values. In some centres, PCD patients also perform multiple breath washout measurements at follow-up, which have been reported to be more sensitive in detecting early lung disease. In those patients, Lung Clearance Index will be considered as a secondary outcome, if patients' numbers allow it. We will use mixed models, including the baseline ENT disease characteristics with an interaction of each with age as fixed effects, and the patients and the centres as random effects.

Finally, we will analyse follow-up data on ENT symptoms and examinations to describe the short-term course of sinonasal and otologic disease during a 2-year follow-up, and identify determinants of disease prognosis. Here, outcomes of interest will be the severity of CRS (based on a symptoms score), loss of smell, bacterial colonisation of the upper respiratory tract, need of surgical management of the sinuses, recurrent otitis media, OME, hearing impairment and need of surgical management of the ears. We will first describe how the outcomes change over the follow-up time in binomial or multinomial (for outcomes with more than three categories, such as hearing impairment) mixed models, using separate models for each outcome. Then, we will study determinants of disease prognosis categorising patients in four groups for each outcome based on the evaluation at baseline and at 2-year follow-up: (a) patients who never reported the outcome, (b) patients negative for the outcome at baseline but positive at follow-up, (c) patients positive at baseline but negative at follow-up and (d) patients positive at both time points. We will use mixed-effects multinomial regression models, including all possible determinants as fixed effects. We will select possible determinants from characteristics available at baseline, including symptoms and measurements; age at diagnosis, ultrastructural defects and genetic mutations; and general patient characteristics, such as age and sex. We will make our determinants selection based on directed acyclic graphs, considering existing literature. Data will be collected prospectively and in a standardised way resulting in small data heterogeneity between participating centres; however, we will still need to account for possible differences between countries in all analyses. Thus, we will analyse all outcomes of interest using random-effects models that account for clustering of patients within participating centres.

For all analyses, results will be interpreted with caution considering the clinical importance and not just mere statistically significant findings. If needed, we will correct for multiple testing. All missing values will be described in detail; however, we expect this will not be an issue for the important variables as the data are collected prospectively. In case we have significant amount of missing data, we will consider subgroup analyses or multiple imputation of missing data to deal with this issue, depending on what is the most appropriate approach for the specific question and analysis.

## ETHICS AND DISSEMINATION
Ethical approval to share pseudoanonymised patient data was obtained locally at each participating centre (online supplemental file). Written informed consent will be obtained in accordance with the national/local data protection laws, from patients aged 14 years or more (might differ per country based on local laws) or from the patient's parents/caregivers for younger patients. Patients can withdraw their consent and their data from the study at any time by contacting the clinical team at each participating centre.

The results of the study will be published in scientific journals and presented at scientific conferences to raise awareness about upper respiratory disease in patients with PCD.

The principal investigator of the study affirms and upholds the principle of the participants' right to dignity, privacy and health and that the project team shall comply with applicable privacy laws. Especially, anonymity of the participants shall be guaranteed when presenting the data at scientific meetings or publishing them in scientific journals. Individual participant medical information obtained during this research project is considered confidential and will not be disclosed to third parties. Data will not be deposited at an open access repository. Because of the rarity of the disease although data will be pseudonymised, it may include still sensitive information; therefore, participants will not be asked to give consent to have their data deposited publicly.

### Patient and public involvement
Several patient support organisations were consulted and support the study. The EPIC-PCD is set up and can be further developed in the framework of the BEAT-PCD clinical research collaboration (CRC) network supported by the ERS.[42] The BEAT-PCD CRC is a large multidisciplinary network, which includes more than 500 researchers and clinicians from more than 30 countries. The EPIC-PCD is part of the BEAT-PCD research portfolio, which ensures widespread dissemination of the study results to researchers, clinicians, patients and all relevant stakeholders. Additionally, the EPIC-PCD study team works closely with a team of patients with PCD who consult BEAT-PCD to ensure that all studies support their interests. Summaries of results in lay language will be disseminated to participants and national patient organisations by the EPIC-PCD data centre and the EPIC-PCD clinics, and to the relevant stakeholders and the public through the BEAT-PCD CRC website and social media.

**Author affiliations**
[1]Institute of Social and Preventive Medicine, University of Bern, Bern, Switzerland

²Paediatric Respiratory Medicine, Children's University Hospital of Bern, University of Bern, Bern, Switzerland
³Hôpital Kremlin-Bicetre, Service d'ORL et de Chirurgie Cervico-Faciale, AP-HP, Paris, France
⁴Faculté de Médecine, Université Paris-Saclay, Le Kremlin-Bicêtre, Paris, France
⁵Department of Otorhinolaryngology, Nicosia General Hospital, Nicosia, Cyprus
⁶Department of Otorhinolaryngology, and Primary Ciliary Dyskinesia Unit, La Fe University and Polytechnic Hospital, Valencia, Spain
⁷Medical School, University of Valencia, Valencia, Spain
⁸Hôpital Henri Mondor et Centre Hospitalier Intercommunal de Créteil, Service d'Oto-Rhino-Laryngologie et de Chirurgie Cervico-Faciale, AP-HP, Creteil, France
⁹Department of Paediatrics, University Hospitals Leuven, Leuven, Belgium
¹⁰Primary Ciliary Dyskinesia Centre, Southampton Children's Hospital, Southampton NHS Foundation Trust, Southampton, UK
¹¹Faculté de médecine, Institut National de la Santé et de la Recherche Médicale, Centre National de la Recherche Scientifique, Hôpital Henri Mondor, Université Paris-Est Créteil Val de Marne, Creteil, France
¹²Department of Pediatric Pulmonology, Faculty of Medicine, Hacettepe University, Ankara, Turkey
¹³Department of Pediatric Pulmonology, School of Medicine, Marmara University, Istanbul, Turkey
¹⁴Department of Paediatric Pulmonology, Emma Children's Hospital, Amsterdam UMC, Vrije Universiteit Amsterdam, Amsterdam, The Netherlands
¹⁵Primary Ciliary Dyskinesia Centre, NIHR Respiratory Biomedical Research Centre, University of Southampton, Southampton, UK
¹⁶Department of Otorhinolaryngology, Head and Neck Surgery, University Hospital of Bern, University of Bern, Bern, Switzerland
¹⁷Department of Respiratory Diseases, University Hospitals Leuven, Leuven, Belgium
¹⁸Biomedical Sciences Department, CEU Cardenal Herrera University, Castellón, Spain
¹⁹Molecular, Cellular and Genomic Biomedicine Group, IIS La Fe, Valencia, Spain
²⁰Department of Pediatric Pulmonology, Immunology and Critical Care Medicine, Charité – Universitätsmedizin Berlin, Berlin, Germany
²¹Department of Otolaryngology, Head and Neck Surgery, Amsterdam University Medical Centres, Amsterdam, The Netherlands
²²Medical School, University of Cyprus, Nicosia, Cyprus
²³Pediatric Pulmonology Unit, Hospital 'Archbishop Makarios III', Nicosia, Cyprus
²⁴Department of Otorhinolaryngology, Head and Neck Surgery, Charité – Universitätsmedizin Berlin, Berlin, Germany

**Acknowledgements** We thank all the patients with primary ciliary dyskinesia (PCD) in the cohort and their families, and the PCD support organisations (especially, PCD Family Support Group UK; Association ADCP France; Kartagener Syndrom und Primäre Ciliäre Dyskinesie e. V. Deutschland/ Deutschschweiz) for their close collaboration. We also thank all the researchers in the participating centres who are involved in recruitment, data collection and data entry, and work closely with us through the whole process of participating to the ear–nose–throat prospective international cohort of patients with primary ciliary dyskinesia.

**Collaborators** For the ear–nose–throat prospective international cohort of patients with primary ciliary dyskinesia (EPIC-PCD) team (listed in alphabetical order): Dilber Ademhan (Hacettepe University, Turkey), Mihaela Alexandru (AP-HP, France), Andreas Anagiotos (Nicosia General Hospital, Cyprus), Miguel Armengot (La Fe University and Polytechnic Hospital, Spain), Emilie Bequignon (AP-HP, France), Irma Bon (Vrije Universiteit, the Netherlands), Mieke Boon (University Hospital Leuven, Belgium), Marina Bullo (University of Bern, Switzerland), Andrea Burgess (University of Southampton, UK), Carmen Casaulta (University of Bern, Switzerland), Marco Caversaccio (University of Bern, Switzerland), Nathalie Caversaccio (University of Bern, Switzerland), Andre Coste (AP-HP, France), Bruno Crestani (RESPIRARE, France), Sandra Diepenhorst (Vrije Universiteit, The Netherlands), Nagehan Emiralioglu (Hacettepe University, Turkey), Ela Erdem (Marmara University, Turkey), Pinar Ergenekon (Marmara University, Turkey), Nathalie Feyaerts (University Hospital Leuven, Belgium), Gabriel Georgiou (Nicosia General Hospital, Cyprus), Amy Glen (University of Southampton, UK), Christine van Gogh (Vrije Universiteit, the Netherlands), Yasemin Gokdemir (Marmara University, Turkey), Myrofora Goutaki (University of Bern, Switzerland), Onder Gunaydın (Hacettepe University, Turkey), Eric G Haarman (Vrije Universiteit, The Netherlands), Amanda Harris (University of Southampton, UK), Sara-Lynn Hool (University of Bern, Switzerland), Isabelle Honoré (RESPIRARE, France), Hasnaa Ismail-Koch (University of Southampton, UK), Bulent Karadag (Marmara University, Turkey), Elisabeth Kieninger (University of Bern, Switzerland), Sookyung Kim (AP-HP, France), Panayiotis Kouis (University of Cyprus, Cyprus), Bernard Maitre (RESPIRARE, France), David Montani (RESPIRARE, France), Yin Ting Lam (University of Bern, Switzerland), Philipp Latzin (University of Bern, Switzerland), Marie Legendre (RESPIRARE, France), Natalie Lorent (University Hospital Leuven, Belgium), Jane S Lucas (University of Southampton, UK), Alison McEvoy (University of Southampton, UK), Rana Mitri-Frangieh (RESPIRARE, France), Loretta Müller (University of Bern, Switzerland), Noelia Muñoz (La Fe University and Polytechnic Hospital, Spain), Ugur Ozcelik (Hacettepe University, Turkey), Beste Ozsezen (Hacettepe University, Turkey), Samantha Packham (University of Southampton, UK), Jean-François Papon (AP-HP, France), Ana Reula (La Fe University, Spain), Jobst Roehmel (Charité-Universitätsmedizin Berlin, Germany), Simone Tanner (Vrije Universiteit, the Netherlands), Guillaume Thouvenin (RESPIRARE, France), Woolf T Walker (University of Southampton, UK), Hannah Wilkins (University of Southampton, UK), Panayiotis Yiallouros (University of Cyprus, Cyprus), Soeren Marian Zappe (Charité-Universitätsmedizin Berlin, Germany).

**Contributors** MG conceptualised and designed the study. MG, YTL and JFP drafted the protocol. YTL, MAlexandru, AA, MArmengot, EB, MB, AB, AC, NE, EE, EGH, AH, S-LH, BK, SK, PL, NL, UO, AR, JR, CvG, PY, SMZ and JFP are responsible for the study conduct and data acquisition. MG and YTL are responsible for data analysis. MG, YTL, MAlexandru, AA, MArmengot, EB, MB, AB, AC, NE, EE, EGH, AH, S-LH, BK, SK, PL, NL, UO, AR, JR, CvG, PY, SMZ and JFP are responsible for the data interpretation and have critically revised and approved this manuscript.

**Funding** This work was supported by the Swiss National Science Foundation grant number (SNFPZ00P3_185923). Future grant applications by all partners to different funding bodies will provide continuous support. Authors participate in the better experimental approaches to treat primary ciliary dyskinesia (PCD) clinical research collaboration, supported by the European Respiratory Society, and most participating centres are part of the European Reference Network-LUNG PCD-Core.

**Map disclaimer** The inclusion of any map (including the depiction of any boundaries therein), or of any geographic or locational reference, does not imply the expression of any opinion whatsoever on the part of BMJ concerning the legal status of any country, territory, jurisdiction or area or of its authorities. Any such expression remains solely that of the relevant source and is not endorsed by BMJ. Maps are provided without any warranty of any kind, either express or implied.

**Competing interests** PL reports personal fees from Gilead, Novartis, OM pharma, Polyphor, Roche, Santhera, Schwabe, Vertex, Vifor and Zambon, and grants from Vertex, all outside the submitted work.

**Patient consent for publication** Not applicable.

**Provenance and peer review** Not commissioned; externally peer reviewed.

**Data availability statement** Data are available upon reasonable request.

**ORCID iDs**
Myrofora Goutaki http://orcid.org/0000-0001-8036-2092
Yin Ting Lam http://orcid.org/0000-0002-2380-834X

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
