## [Reviewer comments · BMJ Open]

ARTICLE DETAILS

TITLE (PROVISIONAL)	Study protocol: the Ear-Nose-Throat (ENT) Prospective International Cohort of patients with Primary Ciliary dyskinesia (EPIC-PCD)
AUTHORS	Goutaki, Myrofora; Lam, Yin Ting; Alexandru, Mihaela; Anagiotos, Andreas; Armengot, Miguel; Bequignon, Emilie; Boon, Mieke; Burgess, Andrea; Coste, Andre; Emiralioglu, Nagehan; Erdem, Ela; Haarman, Eric G.; Harris, Amanda; Hool, Sara-Lynn; Karadag, Bulent; Kim, Sookyung; Latzin, Philipp; Lorent, Natalie; Ozcelik, Ugur; Reula, Ana; Roehmel, Jobst; van Gogh, Christine; Yiallourous, Panayiotis; Zappe, Soeren; PAPON, Jean

VERSION 1 – REVIEW

REVIEWER	Lourencone, Luiz University of Sao Paulo, Otolaryngology
REVIEW RETURNED	16-May-2021

GENERAL COMMENTS	The manuscript has a relevant objective and its method is well designed, and with the expectation of a representative multicenter sample. The standard of the exam, in this prospective study, could be better defined among which findings they will look for in nasal endoscopy, for example, so that all centers value the same clinical findings (quantify secretion and location, for example). The realization of a well-established Protocol is, without a doubt, an important step for the relevant progression of this manuscript. Despite the excellent expectation with the result of this manuscript, I recommend that the editor evaluate whether the protocol proposal is recognized to justify publication in its important journal.
---

REVIEWER	Zawawi, Faisal King Abdulaziz University, Otolaryngology - Head & Neck Surgery
REVIEW RETURNED	03-Aug-2021

GENERAL COMMENTS	I would like to commend the authors for tackling such an important topic. The literature is still scarce when it comes to upper airway disease in patients with PCD. Never the less I would strongly recommend that the authors review some the recently published articles in the field including the newly published cross-sectional multicenter study looking at the Otolaryngology related manifestations of PCD in children. This might help fine tune the protocol further. Zawawi F, Shapiro AJ, Dell S, Wolter NE, Marchica CL, Knowles
---

	MR, Zariwala MA, Leigh MW, Smith M, Gajardo P, Daniel SJ. Otolaryngology Manifestations of Primary Ciliary Dyskinesia: A Multicenter Study. Otolaryngol Head Neck Surg. 2021 Jun 22:1945998211019320. doi: 10.1177/01945998211019320. Epub ahead of print. PMID: 34154450. - I would also standardize the clinical assessment protocol. Which means, every patient would be asked the exact same questions in history, and the examination is reported in a standardized form (e.g. Using Brodsky's tonsil size or Meltzer polyps grading system). - Please consider attaching the actual clinical assessment tools to the protocol to help interested parties join if they wish to. - The tests & questionnaires mentioned on the protocol seem a bit loose in requirements. I would strongly recommend making these tests either mandatory or removing them from the protocol. This will give strengths to the data. - Diagnostic Information: I would recommend highlighting in the protocol the PCD diagnostic criteria, e.g. Genetic or phenotype. Please refer to the PCD Clinic Practice Guidelines Shapiro AJ, Davis SD, Polineni D, Manion M, Rosenfeld M, Dell SD, Chilvers MA, Ferkol TW, Zariwala MA, Sagel SD, Josephson M, Morgan L, Yilmaz O, Olivier KN, Milla C, Pittman JE, Daniels MLA, Jones MH, Janahi IA, Ware SM, Daniel SJ, Cooper ML, Nogee LM, Anton B, Eastvold T, Ehrne L, Guadagno E, Knowles MR, Leigh MW, Lavergne V; American Thoracic Society Assembly on Pediatrics. Diagnosis of Primary Ciliary Dyskinesia. An Official American Thoracic Society Clinical Practice Guideline. Am J Respir Crit Care Med. 2018 Jun 15;197(12):e24-e39. doi: 10.1164/rccm.201805-0819ST. PMID: 29905515; PMCID: PMC6006411. - If questionnaires, or tests/microbiology, to be used I would highly recommend to have the timing standardized. This will give the most meaning to the data. For an exam, if you are to use SNOT-22 you can say, SNOT-22 at enrolment to the study, and SNOT-22 at 6 months post enrolment. This should be applied to all tests.
--	--

VERSION 1 – AUTHOR RESPONSE

Reviewer Reports:

Reviewer: 1

Dr. Luiz Lourencone, University of Sao Paulo

Comments to the Author:

The manuscript has a relevant objective and its method is well designed, and with the expectation of a representative multicenter sample.

The standard of the exam, in this prospective study, could be better defined among which findings they will look for in nasal endoscopy, for example, so that all centers value the same clinical findings (quantify secretion and location, for example).

The realization of a well-established Protocol is, without a doubt, an important step for the relevant progression of this manuscript.

Reply: We would like to thank the reviewer for this positive review and his comments. Using the standardised FOLLOW-PCD form, we have already defined which outcomes we will collect and evaluate for the study during the ENT clinical examinations such as nasal endoscopy. We have expanded on this in the section ENT examinations, under “What information is collected». We revised the text adding the following paragraph:

(What information is collected, pages 11-12) The data collected during the ENT examinations are predefined based on the ENT examinations module of the FOLLOW-PCD standardised form. Nasal endoscopy data include presence and consistency of nasal discharge (e.g. serous, muco-purulent), evaluation of nasal mucosa (e.g. presence of erythema or oedema), visual evaluation of nasal polyps using the Lildholdt score, evaluation of hypertrophy or atrophy of nasal turbinates and description of septum and any deformities caused due to chronic infection. Otoscopy data include presence and consistency of ear discharge, visual description of the tympanic membrane (e.g. tympanic perforation, retracted membrane, tympanic sclerosis) and the presence of acute otitis media or OME. Results of tympanometry are recorded using the description of tympanogram type by Jerger and of audiometry using the type of audiometry and the World Health Organization hearing loss grades. In addition, we record the presence of grommets or use of hearing aids. When sinus imaging is performed, we record the following findings: presence of aplasia, hypoplasia, thickening of the bone and Lund-Mackay score.

Reviewer: 2

Dr. Faisal Zawawi, Hospital for Sick Children Research Institute

Comments to the Author:

I would like to commend the authors for tackling such an important topic. The literature is still scarce when it comes to upper airway disease in patients with PCD. Never the less I would strongly recommend that the authors review some the recently published articles in the field including the newly published cross-sectional multicenter study looking at the Otolaryngology related manifestations of PCD in children. This might help fine tune the protocol further.

Zawawi F, Shapiro AJ, Dell S, Wolter NE, Marchica CL, Knowles MR, Zariwala MA, Leigh MW, Smith M, Gajardo P, Daniel SJ. Otolaryngology Manifestations of Primary Ciliary Dyskinesia: A Multicenter Study. *Otolaryngol Head Neck Surg.* 2021 Jun 22:1945998211019320. doi: 10.1177/01945998211019320. Epub ahead of print. PMID: 34154450.

Reply: We would like to thank the reviewer for this positive review and constructive criticism. We agree that the literature on this topic is scarce. We reviewed this recently published study and have added a sentence in the introduction referring to the study:

(Introduction, page 6) In some cases, it may lead to hearing impairment and cause delays in speech development. A newly published study evaluated 47 children in two PCD centres in North America and found that 52% of children with PCD-related hearing loss were not aware of their hearing deficit

that was present on audiological assessment. While otologic involvement seems to improve with age in PCD patients, sinonasal disease appears to persist and may worsen in adulthood.

- I would also standardize the clinical assessment protocol. Which means, every patient would be asked the exact same questions in history, and the examination is reported in a standardized form (e.g. Using Brodsky's tonsil size or Meltzer polyps grading system).

Reply: We would like to thank the reviewer for this comment. As we explain in our response to reviewer 1, we have already pre-defined the clinical assessment protocol using the PCD-specific standardised FOLLOW-PCD form. Specifically, information on clinical history are reported using the FOLLOW-PCD patient questionnaire, which asks the same questions on symptoms to all patients and the examination is reported using the standardised ENT examination module of FOLLOW-PCD (e.g. we use the Lidholdt score for polyps). We have expanded on this topic in the manuscript as follows:

We added a sentence where we describe the FOLLOW-PCD questionnaire in the methods. (Data collection instruments, page 10) Some modules of FOLLOW-PCD have been developed into patient questionnaires (FOLLOW-PCD questionnaire) that can be completed by the patients during each clinical follow-up visit. The FOLLOW-PCD questionnaire includes detailed questions on frequency and severity of upper and lower respiratory symptoms and health-related behaviours (e.g. smoking, exercise).

We added a paragraph describing in detail what is reported for the ENT examinations. (What information is collected, pages 11-12) The data collected during the ENT examinations are predefined based on the ENT examinations module of the FOLLOW-PCD standardised form. Nasal endoscopy data include presence and consistency of nasal discharge (e.g. serous, muco-purulent), evaluation of nasal mucosa (e.g. presence of erythema or oedema), visual evaluation of nasal polyps using the Lidholdt score, evaluation of hypertrophy or atrophy of nasal turbinates and description of septum and any deformities caused due to chronic infection. Otoscopy data include presence and consistency of ear discharge, visual description of the tympanic membrane (e.g. tympanic perforation, retracted membrane, tympanic sclerosis) and the presence of acute otitis media or OME. Results of tympanometry are recorded using the description of tympanogram type by Jerger and of audiometry using the type of audiometry and the World Health Organization hearing loss grades. In addition, we record the presence of grommets or use of hearing aids. When sinus imaging is performed, we record the following findings: presence of aplasia, hypoplasia, thickening of the bone and Lund-Mackay score.

Please consider attaching the actual clinical assessment tools to the protocol to help interested parties join if they wish to.

Reply: The assessment tools used for the clinical examination as well as the patient questionnaires on symptoms, lifestyle and quality of life are already described in detail in previous publications for FOLLOW-PCD and QoL-PCD (Goutaki et al. ERJ Open Res. 2020 Feb 10;6(1):00237-2019 and Lucas et al. Eur Respir J. 2015 Aug;46(2):375-83).

These instruments are copyrighted and used in several studies therefore we cannot attach them with this protocol, however as we explained in the previous point, we have revised the text to clarify which exact information we collect during the clinical visit. Any interested parties could access the documents after communication by contacting the study team.

- The tests & questionnaires mentioned on the protocol seem a bit loose in requirements. I would strongly recommend making these tests either mandatory or removing them from the protocol. This will give strengths to the data.

Reply: We regret if the requirements for study participation were unclear. This study is embedded in regular clinical care and the examinations and tests are therefore performed as part of the regular patient follow-up and not just for the study. However, there are clear eligibility criteria for the participation. These are summarized in figure 2 as minimum requirements at baseline and at follow-up visits. Mandatory tests that will be performed by all patients include nasal endoscopy, audiometry and tympanometry as well as spirometry. These data together with information on symptoms collected from all participants using the FOLLOW-PCD patient questionnaire will be the focus of the study. However, since the study offers an excellent opportunity to collect information from additional examinations performed routinely only in few centres (e.g. multiple breath washout) for relevant subgroup analysis. In addition to figure 2 where this distinction is made, we revised the text to clarify this issue and highlight the mandatory requirements of the study.

(What information is collected, page 11) Figure 2 provides a brief overview of the information collected from participating patients at the baseline and follow-up visits, including the minimum required data, which are assessed for all patients in all participating centres and additional data from tests performed in few centres and which will be collected if available for subgroup analysis to address additional research questions. Small exemptions are allowed when a minimally required examination was not performed due to e.g. the patient's very young age, as long as each patient undergoes a specialised ENT examination at least once a year.

- Diagnostic Information: I would recommend highlighting in the protocol the PCD diagnostic criteria, e.g. Genetic or phenotype. Please refer to the PCD Clinic Practice Guidelines Shapiro AJ, Davis SD, Polineni D, Manion M, Rosenfeld M, Dell SD, Chilvers MA, Ferkol TW, Zariwala MA, Sagel SD, Josephson M, Morgan L, Yilmaz O, Olivier KN, Milla C, Pittman JE, Daniels MLA, Jones MH, Janahi IA, Ware SM, Daniel SJ, Cooper ML, Nogee LM, Anton B, Eastvold T, Ehrne L, Guadagno E, Knowles MR, Leigh MW, Lavergne V; American Thoracic Society Assembly on Pediatrics. Diagnosis of Primary Ciliary Dyskinesia. An Official American Thoracic Society Clinical Practice Guideline. *Am J Respir Crit Care Med.* 2018 Jun 15;197(12):e24-e39. doi: 10.1164/rccm.201805-0819ST. PMID: 29905515; PMCID: PMC6006411.

Reply: The PCD diagnostic criteria used for this study are based on the European Respiratory Society diagnostic guidelines as these represent the diagnostic procedures followed and the tests performed at the participating centres, particularly due to differences in tests expertise and availability (e.g. for high speed videomicroscopy).

Lucas JS, Barbato A, Collins SA, et al. European Respiratory Society guidelines for the diagnosis of primary ciliary dyskinesia. *Eur Respir J.* 2017 Jan 4;49(1):1601090. doi: 10.1183/13993003.01090-2016.

We already refer to the ERS diagnostic guidelines in the study population section (inclusion criteria page 8): Patients of all ages, diagnosed with PCD, according to the European Respiratory Society (ERS) guidelines, and followed-up at one of the participating centres are invited to participate to the EPIC-PCD.

- If questionnaires, or tests/microbiology, to be used I would highly recommend to have the timing standardized. This will give the most meaning to the data. For an exam, if you are to use SNOT-22 you can say, SNOT-22 at enrolment to the study, and SNOT-22 at 6 months post enrolment. This should be applied to all tests.

Reply: We would like to thank the reviewer for this suggestion. Since the purpose of the study is to be embedded in regular care, the timing of the examinations (questionnaires and tests performed) is depended on the follow-up intervals for each patient at the centre. This is extremely important for observational studies particularly since the beginning of the pandemic that clinical follow-up of patients had to be adjusted to the regulations and research projects that were not embedded in regular care almost entirely stopped.

In most countries, visits to the PCD centre are more regular for paediatric patients (e.g. every 3 months). Adult patients tend to only have a 6-monthly or annual follow-up visit. For this reason, all patients complete the FOLLOW-PCD questionnaire and the mandatory tests at enrolment and every 12 months. In addition, if a patient has an earlier visit, e.g. at 3 or 6 months, we collect information on tests using the standardised form, and the patient is asked to complete the questionnaire. This design allows for standardised yearly timing as suggested by the reviewer, increased participation of patients and clinicians but also richness of data recorded at all visits.